# Multifunctional Textile Platform for Fiber Optic Wearable Temperature-Monitoring Application

**DOI:** 10.3390/mi10120866

**Published:** 2019-12-10

**Authors:** Ziyang Xiang, Liuwei Wan, Zidan Gong, Zhuxin Zhou, Zhengyi Ma, Xia OuYang, Zijian He, Chi Chiu Chan

**Affiliations:** 1Key Laboratory of Advanced Optical Precision Manufacturing Technology of Guangdong Higher Education Institutes, Sino-German College of Intelligent Manufacturing, Shenzhen Technology University, Shenzhen 518118, China; xiangziyang94@163.com (Z.X.); zhouzhuxin1219@163.com (Z.Z.); mazhengyi876@163.com (Z.M.); chenzhichao@sztu.edu.cn (C.C.C.); 2College of Optical and Electronic Technology, China Jiliang University, Hangzhou 310018, China; wanliuwei@sztu.edu.cn; 3Department of Electrical Engineering, The Hong Kong Polytechnic University, Hong Kong, China; xia.ouyang@connect.polyu.hk (X.O.); 16097363d@connect.polyu.hk (Z.H.)

**Keywords:** textile platform, fiber optic, wearable application, temperature monitoring

## Abstract

Wearable sensing technologies have been developed rapidly in the last decades for physiological and biomechanical signal monitoring. Much attention has been paid to functions of wearable applications, but comfort parameters have been overlooked. This research presents a developed fabric temperature sensor by adopting fiber Bragg grating (FBG) sensors and processing via a textile platform. This FBG-based quasi-distributed sensing system demonstrated a sensitivity of 10.61 ± 0.08 pm/°C with high stability in various temperature environments. No obvious wavelength shift occurred under the curvatures varying from 0 to 50.48 m^−1^ and in different integration methods with textiles. The temperature distribution monitored by the developed textile sensor in a complex environment with multiple heat sources was deduced using MATLAB to present a real-time dynamic temperature distribution in the wearing environment. This novel fabric temperature sensor shows high sensitivity, stability, and usability with comfort textile properties that are of great potential in wearable applications.

## 1. Introduction

Technological elements can be incorporated into fabrics at any level to fabricate smart textile materials where fabrics have been treated or modified to act as sensors, actuators, and/or other types of transducers [1]. Taking into account flexible structure design, diverse fabrication techniques, excellent comfort performance, and biocompatibility of textiles, a multifunctional textile platform can then be established by adopting various highly advanced technologies for wearable application development. For instance, electrotextiles for wearables with antistatic function, infrared absorption, and explosive area protection have been developed using metal monofilaments (e.g., Cu/Ag, Ms/Ag, and Al filaments) as conductive fibers in weaving and knitting textile fabrication techniques [2,3]. Biomaterials such as plant fiber and biological micro/nanostructures such as the “shark skin effect” and “lotus effect” have already been applied in textile structures or surfaces to develop high-performance textiles with desired functions (e.g., drag reduction, self-cleaning, microbe resistant, environmental friendly) [4,5,6]; flexible and soft sensors were integrated into wearable devices to set up textile-based non-invasive monitoring systems which could be easily attached to the human body for physical and vital signal detecting [7,8,9].

Fiber optic technology is gaining wide acceptance due to its good metrological properties and material characteristics. The fibrous appearance of optical fibers is an advantage for application in textile-based sensors because optical fibers are quite similar to traditional textile fibers that could be ideally processed as standard textile yarns for fabric fabrication and thus produce smart and functional textile materials. In this case, the functions of fiber optic technology could reach any part of the curved human body via a textile-based wearable modality such as tights [10,11]. Moreover, compared with electronic devices, optical fiber sensors have various advantages: they are durable, immune to electromagnetic interference, exhibit resistance to salt solutions (i.e., sweat), and possess good biocompatibility with the human body [10,11,12,13]. These unique features make optical fiber sensors particularly suitable for special environments, i.e., MRI environments and the skin surface [12]. Among numerous fiber optic sensors, fiber Bragg grating (FBG) sensors are being developed to become a prominent technology and are commercialized in a broad field of applications. Hill and his colleagues first demonstrated the formation of permanent grating in optical fibers in 1978 at the Canadian Communications Research Centre (Ottawa, ON, Canada). Argon ion laser radiation was launched in a germania-doped fiber on a short piece and, thereby, the intensity of the reflected light could be monitored [13,14]. Until the 2000s, FBG sensors were maturely fabricated into standard optical sensors by using various techniques, including phase mask, pico-, and femtosecond laser inscription, and interferometric setting [15,16,17]. Recently, Bragg grating fabrication based on various materials or special fiber structure design is emerging rapidly [18,19,20,21], for instance, a chirped Bragg grating on microstructured polymethyl methacrylate fiber proposed by Korganbayev et al. (2018) [22] that demonstrated excellent sensitivity for thermal profile detection; and some multicore fibers (MCFs) have been used as fiber optic three-dimensional shape sensors such as the helical MCF with continuous grating developed by Lally et al. (2012) [23] for real-time shape sensing using the optical frequency domain reflectometry technique. Such sensors have been widely applied in civil engineering [24], healthcare medical devices [25,26], structural engineering [27,28], aerospace [29], oil and fuels [30], and harsh environments [31], among others.

Temperature is a commonly tested physical parameter by FBG sensors closely associated with healthcare management, medical applications, fire safety, and environmental detection [32,33,34,35]. This study aims to adopt FBG sensors in a multifunctional textile platform to develop a textile-based quasi-distributed wearable temperature-monitoring system consisting of multiple FBGs in serial connection. This developed system with testing points or networks creates a fabric sensor that is sensitive to real-time and dynamic temperature distributions. The system also exhibits excellent stability under a wide range of working temperatures. A novel and intelligent fabric material that can be widely applied in wearable temperature-monitoring applications was then designed.

## 2. Materials and Methods

### 2.1. Working Principle of the FBG-Based Fabric Sensor for Temperature Monitoring

Figure 1 demonstrates the working mechanism of the FBG-based fabric sensor. The FBG temperature sensor was embedded into textiles by different textile fabrication techniques via the textile platform to produce a fabric sensor. FBG is a distributed Bragg reflector that is constructed in short segments of optic fiber that reflects a particular wavelength of transmitted light. This is achieved by creating a periodic variation in the refractive index of the fiber core.

The Bragg wavelength $\lambda_{B}$ or the strongest reflected spectral signal is given by the following equation [36]:
(1)$$\lambda_{B}=2n_{eff}\Lambda$$
where $n_{eff}$ is the effective refractive index of the core, and Λ is the grating pitch. Thus, the Bragg wavelength can be shifted by changing the refractive index $n_{eff}$ or the grating pitch Λ. When external perturbation, such as temperature and strain, is applied to the FBG, as illustrated in Figure 1, the effective refractive index or grating pitch will change. The reflected Bragg wavelength will shift from $\lambda_{B}$ to ${\lambda_{B}}^{\prime }$ in accordance with Equation (1), and the variation of the reflected spectrum will be detected. The measured strain responsivity at constant temperature can be expressed as
(2)$$\frac{1}{\lambda_{B}}\frac{\delta \lambda_{B}}{\delta_{\varepsilon }}=0.78\times 10^{-6}\;{\mu \varepsilon }^{-1}.$$


The measured temperature response at constant strain can be expressed as Equation (3). For the scope of this study, the conducted investigations focus only on temperature.
(3)$$\frac{1}{\lambda_{B}}\frac{\delta \lambda_{B}}{\delta T}=6.67\times 10^{-6}\;{{}^{{}^{\circ}}C}^{-1}$$


### 2.2. Implementation of the Experiments

Experiments were conducted in a closed room with small air flow with a room temperature of 20 °C (±0.5 °C). The effects of temperature on the FBG system were investigated to develop the FBG-based quasi-distributed wearable temperature-monitoring system via the textile platform. An amplified spontaneous emission (ASE) broad-spectrum light source (ASE-50 mW, Shenzhen Hoyatek Co., Ltd., Shenzhen, China; wavelength: 1527–1565 nm; output power: 17 dBm) was adopted, and the spectral characteristics were detected by an optical spectrum analyzer (OSA, AQ6370D, Yokogawa Ltd., Tokyo, Japan) with a maximum wavelength resolution of 0.02 nm. A programmable temperature and humidity chamber (TEMIP950, Mingjia Co., Ltd., Hangzhou, China) was used as an experimental thermal chamber to simulate the varying temperature environment allowing for FBG sensors working in different temperature levels. The temperature uniformity of the thermal chamber was ±2 °C, and the temperature fluctuation was ±0.5 °C. A pair of fiber holders (xyz60, Beijing Optical Century Instrument Co., Ltd., Beijing, China) were used for fiber curvature measurement to simulate fiber bending in practical use, and digital hot plates (PC-420D, Corning Incorporated, Corning, NY, USA) were adopted to simulate the complex temperature situation with multiple heat sources. FBG-based quasi-distributed sensors ($\lambda_{B1}$ = 1533.6 nm, $\lambda_{B2}$= 1538.4 nm, $\lambda_{B3}$ = 1545.6 nm, $\lambda_{B4}$ = 1552.8 nm) in serial connection were prepared to connect to a 3 dB coupler, which was in connection with the ASE and OSA. The light source produced by ASE was transferred to the FBG sensor via the coupler, and the temperature variations sensed by FBG were then delivered to the OSA. The experiment comprised four main parts, as demonstrated in Figure 2b–e.

The FBG sensors in serial connection were first exposed to the same conditions in the controlling thermal chamber for temperature variation monitoring in order to explore the sensitivity and stability of the developed quasi-distributed temperature-monitoring system, as presented in Figure 2b. The first setting temperature was 20 °C, the last setting temperature was 130 °C, and the heating distant temperature was 10 °C. This procedure was repeated three times. The wavelength of each FBG sensors was recorded at each temperature interval, and no thermal hysteresis occurred during data acquisition. The entire system was then placed in constant-temperature conditions of 40, 80, and 120 °C and maintained for 1 h for wavelength shift monitoring every 5 min.

Figure 2c depicts the fiber holder setup to monitor FBG central wavelength variations under different curvature situations. Two ends of the sensing area were placed on the fiber holder with an initial distance of 45.5 mm, and the separation distance between the two holders was adjusted accurately by moving one holder inward every 0.5 mm each time. In this way, the fiber curvature (C) can be expressed as [37,38,39]
(4)$$C=1/R≅\sqrt{24x/L_{0}^{3}}$$
where *R* is the radius of the bent fiber, $L_{0}$ = 45.5 mm is the distance between the two fiber holders when the fiber was straightened, and *x* is the inward distance of the moving holder. When the sensing area was bent, potential central wavelength shifts of the FBG were observed and recorded.

A textile structure of hollow double-wall fabric was adopted as a base, where the quasi-distributed FBG sensors were embedded by the methods of between-walls and cross-walls for intelligent fabric sensor development (Figure 2d). FBGs in different textile integration methods were placed in the temperature chamber for a sensing capability test. The chamber was programmed to increase the temperature from 20 to 130 °C with 10 °C steps and then decrease back to 20 °C with the same procedure. The heating and cooling procedures were repeated three times, and all spectrum data at each temperature were recorded by the OSA to demonstrate the consistency of the sensors.

Finally, a complex practical temperature environment with multiple heat sources of 40, 60, 80, and 100 °C was simulated by digital hot plates to test the real-time temperature distribution of the developed fabric sensor, as shown in Figure 2e. Four FBGs with different central wavelengths integrated in textile were placed on the hot plates of various temperatures separately, and the central wavelengths of the different gratings were collected every 5 min and analyzed by the OSA. Simulations were conducted at this stage in order to verify the experimental results and demonstrate the dynamic thermal distribution on the fabric sensor in practical use monitored by a certain number of FBG sensors.

## 3. Results and Discussion

Four quasi-distributed FBG sensors with different initial wavelengths (i.e., $\lambda_{B1}$= 1533.6 nm, $\lambda_{B2}$= 1538.4 nm, $\lambda_{B3}$ = 1545.6 nm, $\lambda_{B4}$ = 1552.8 nm) were tested for wavelength shift in the temperature-controlling thermal chamber. The results are presented in Figure 3 below. Redshift (wavelength increase) occurred in all of the FBGs during each recording interval (every 10 °C) from 20 to 130 °C. Table 1 summarizes the results of the thrice-repeated test, which indicated the good sensitivity consistency of the developed monitoring system. Figure 3a–d presents the wavelength shift and corresponding spectrums of FBGs 1–4 along with the temperature variation. To obtain a comprehensive linear characteristic of the FBG-based quasi-distributed monitoring system, an integrated fitting curve is presented in Figure 3e which indicates a stable sensitivity of 10.61 ± 0.08 pm/°C.

The stability of the quasi-distributed FBG system in various constant temperature conditions was analyzed, as presented in Figure 4. No significant wavelength variation was observed in each FBG within the setting period (1 h) under certain temperatures (40, 80, and 100 °C), as illustrated in Figure 4a. Figure 4b shows that the wavelength drift of all FBGs at different temperature conditions was within 0.01 nm on average; even the maximum wavelength drift was under 0.025 nm. The general temperature sensitivity of the FBG was 10 pm/°C, so an error range of ±1 °C could be controlled, indicating good temperature stability in practical use.

Figure 5 shows the central wavelength variation with changing bending curvature. When the curvature varied from 0 to 50.48 m^−1^, no obvious wavelength shift occurred, presenting excellent sensing stability in bending situations. The bending response of the FBG sensor depended on the spatial position relationship between the core region and symmetry center of the fiber. When the FBG core region was out of the center of symmetry, the working area was likely to experience extension or compression during bending, leading to a wavelength shift. Kong et al. conducted a bending experiment on FBG at the connection joint of eccentric-core and single-mode fibers with the results of redshift and blueshift representing near-linear features [40]. Chen et al. studied the bending impact on the Bragg grating written in an eccentric-cored polymer optical fiber, with the results indicating a strong orientational dependence and high bend sensitivity [41]. Osorio et al. explored the bending impact on a surface-core fiber and observed a directional behavior of the optical response in the curvature probing due to the fiber geometry [42]. When the core lay on the center axis, as represented in the results of the study of Kong et al., no obvious wavelength shift was observed in the test, which was confirmed by our curvature testing results [40]. This finding can be explained by the circularly symmetrical structure of the single-mode fiber, in that bending applied on any direction of the fiber will not cause much extension or compression to the core. With regards to longitudinal strain applied to the fiber, from the scope of current study, because of the loose optic fiber integration method into textiles, the optimized textile platform minimized the longitudinal strain applied on the FBG sensor. Considering the longitudinal strain impact on FBG sensors in some wearable applications, typically values for the sensitivity to an applied longitudinal strain would be 1 pm/με and our proposed thermal sensitivity is 10.61 ± 0.08 pm/°C; therefore a small theoretical cross-sensitivity of 0.094 °C/με would be obtained [18]. Thus, the FBG-based quasi-distributed sensing system exhibited excellent sensing stability and could be flexibly applied in different textile structures for various wearable purposes on the curved human body.

In the stage of textile integration, sensitivity of the quasi-distributed FBG system in a hollow double-wall fabric base was tested using the integration methods of between-walls and cross-walls, as presented in Figure 6. The obtained temperature sensitivities of the developed fabric sensor of the two different integration methods were 10.10 and 10.04 pm/°C, respectively. The results demonstrated that the temperature sensitivity of the fabric sensor was not influenced by the FBG integration methods with textiles, which implied good compatibility between textile structure and FBG sensing properties. Moreover, the developed fabric sensor was tested to have light weight (98 g/m^2^), high pilling resistance (level 5), excellent air permeability (0.007 kPa·s^−1^·m^−1^), and high thermal conductivity (Q-Max 0.23 W/cm^2^), which contribute to breathable, lightweight, and durable wearable applications.

On the basis of different application requirements, other textile techniques can be adopted for FBG integration, given different textile materials and structures equipped with various fabric properties for wearable applications. To achieve a thinner handle of developed fabric sensor, optical fibers could be woven into a fabric with a limited bending angle for effective sensing functions, or embedded into multilayer woven structures for a more stable structure with more functions on different layers [43,44,45]. Moreover, a crochet fabrication technique was commonly adopted to integrate optical fibers into textiles with a “sinusoidal” shape [46,47]. Easy stitching and embroidering techniques were usually used to embed optic fibers with fragile structures into fabrics without influencing sensor properties [46,47,48]. Additionally, novel fiber loop configuration can be applied in optic fibers such as the figure-of-eight coil, which results in increased linearity of response, less mechanical resistance and hysteresis, and higher resolution and sensitivity in respiratory monitoring applications [49]. Knitting is another textile technique that can be adopted to develop fabric sensors in the structures of laid-in weft-knitted, laid-in warp-knitted, and single jersey hopsack, etc., with high elasticity [50]. Since FBG sensors have a working area at a certain length, the knitting structure of miss stitch, also known as float stitch, is appropriate for integration because the fiber appears in a U shape as a single yarn floating freely on the fabric material surface (Figure 7). Float numbers determine the floating length, which can be flexibly designed to match the sensing range. This textile structure offers considerable protection for embedded fiber optic technology by avoiding a sharp break on the FBG working area and allowing the sensor to be in full contact with the target temperature environment.

In this study, an 80 cm × 80 cm hollow double-wall fabric sensor was developed for wearable temperature applications, as presented in Figure 8a. The results of the multiple heat source test for real-time temperature detection are illustrated in Figure 8b,c. Four heat sources of 40, 60, 80, and 100 °C were set up on the sensing fabric, matching the four embedded FBGs with the measured temperature distribution shown in Figure 8b and corresponding spectrums in Figure 8c. A high-external-temperature environment induced considerable wavelength shift. However, in practical use, there will be a thermal conducting process surrounding the four heat sources. A certain number of FBGs connected in the quasi-distributed monitoring system was intended to create a sensing network. The thermal distribution of the sensing area applied by external heat sources could then be deduced by the thermal-equation-based model. A model of 80 cm × 80 cm fabric with four evenly distributed given heat sources was adopted with the corresponding thermal equation expressed as
(5)$$\rho C\frac{\partial T}{\partial t}-\nabla \left(k\nabla T\right)=f$$
where *f* is the generated heat, *k* is the thermal conductivity, *C* is the heat capacity, ρ is the fabric density, and *T* is the temperature. Here, *k* is the thermal conductivity of air and is regarded as a straight line with respect to temperature. Only a steady-state solution was considered, and ρ, *C*, and initial conditions were not specified. *f* was 0, given that the characteristics of heat sources were indicated by temperature instead of power. The boundary condition of the sample was room temperature (20 °C), and the outlines of the heat sources were 40, 60, 80, and 100 °C. According to simulation, the time duration for a 99% level of steady temperature at the center point is around 45 s. Considering the experiment time interval of 5 min, that is long enough to get accurate and stable data for the simulation of temperature distribution of the sample. Moreover, the time response of silica optical fiber FBG with respect to temperature change is typically less than 1 s [51], which is much less than the time interval of the experiment. Therefore, FBG thermal sensors can be used in predicting thermal distribution. The simulation results are explicitly displayed as 3D and 2D in Figure 8d. The results verified the experimental outcomes and deduced the dynamic thermal distribution of the intelligent wearable fabric sensor with a certain number of FBG sensors embedded, which could be applied in a complex temperature environment.

## 4. Conclusions

In this study, a novel and intelligent fabric sensor for temperature monitoring was developed via the textile platform, and its sensitivity, stability, and usability in wearable application were demonstrated by a series of experiments. This FBG-based quasi-distributed sensing system presented a high sensitivity of 10.61 ± 0.08 pm/°C, with high stability and consistency in various temperature environments. No obvious wavelength shift was observed in the bending test, implying excellent usability of the developed system in a fabric sensor that could be widely applied in wearable applications. Simulations were conducted to verify the experimental results and deduce the dynamic gradient thermal distribution on the fabric sensor in practical use, monitored by a sensing network which was created by a certain number of FBG sensors. As people prefer to wear comfortable textiles rather than hard and inelastic devices directly attached to the human skin, this monitoring system was reprocessed in a textile platform. The developed textile-based FBG quasi-distributed system not only features a real-time dynamic temperature-detecting function but also offers textile properties, such as being lightweight, elastic, breathable, thermally conductive, and pilling resistant. The system can also be equipped with other functions (e.g., waterproof or antibacterial) by integrating it with high-performance and multifunctional textiles. The multifunctional textile platform is a promising wearable carrier for optic fibers and other technologies for interdisciplinary application.

## Figures and Tables

**Figure 1 micromachines-10-00866-f001:**
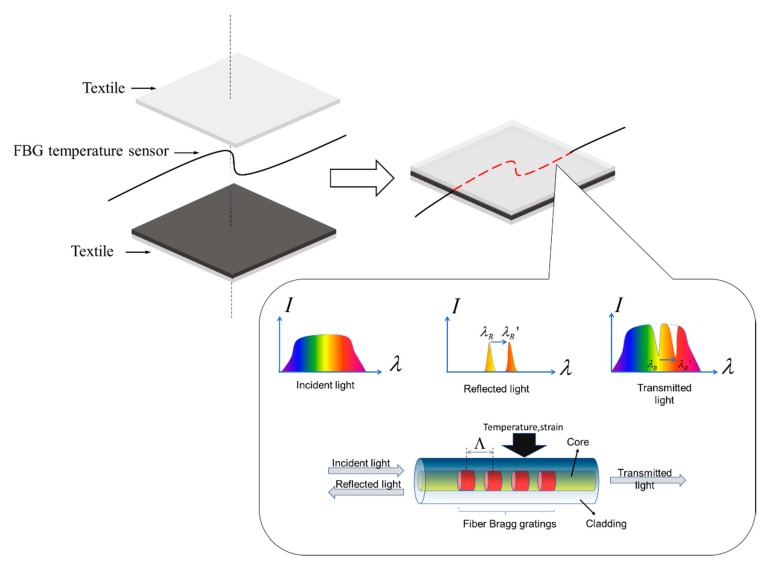
Working mechanism of the fiber Bragg grating (FBG).

**Figure 2 micromachines-10-00866-f002:**
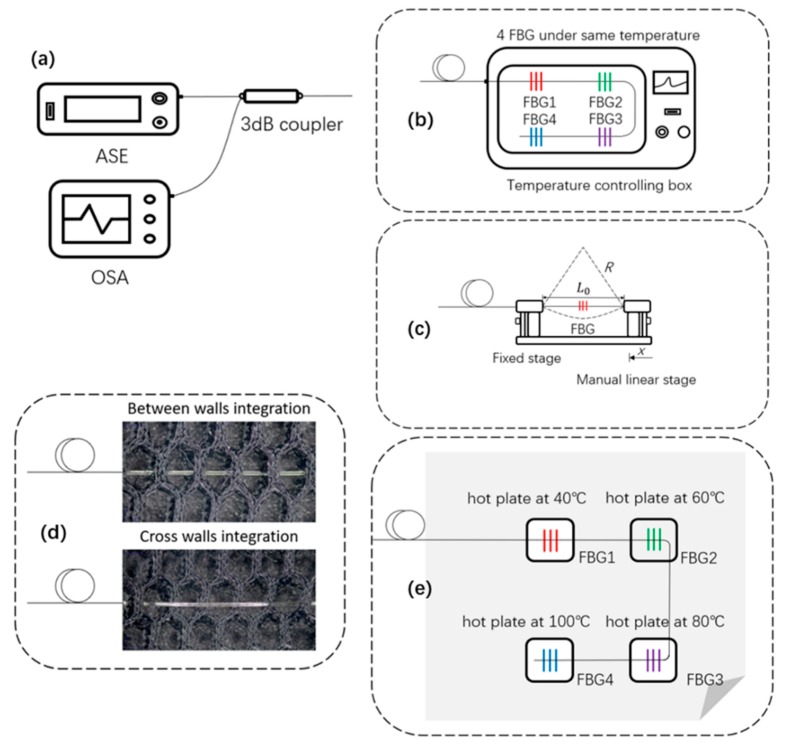
Schematic of the experiment implementation: (**a**) basic test connecting equipment, (**b**) sensitivity and stability test in controlling thermal chamber, (**c**) curvature measurement on fiber holders, (**d**) fabric temperature sensor capability test, (**e**) fabric temperature sensor for multiple heat source simulation. ASE: amplified spontaneous emission; OSA: optical spectrum analyzer.

**Figure 3 micromachines-10-00866-f003:**
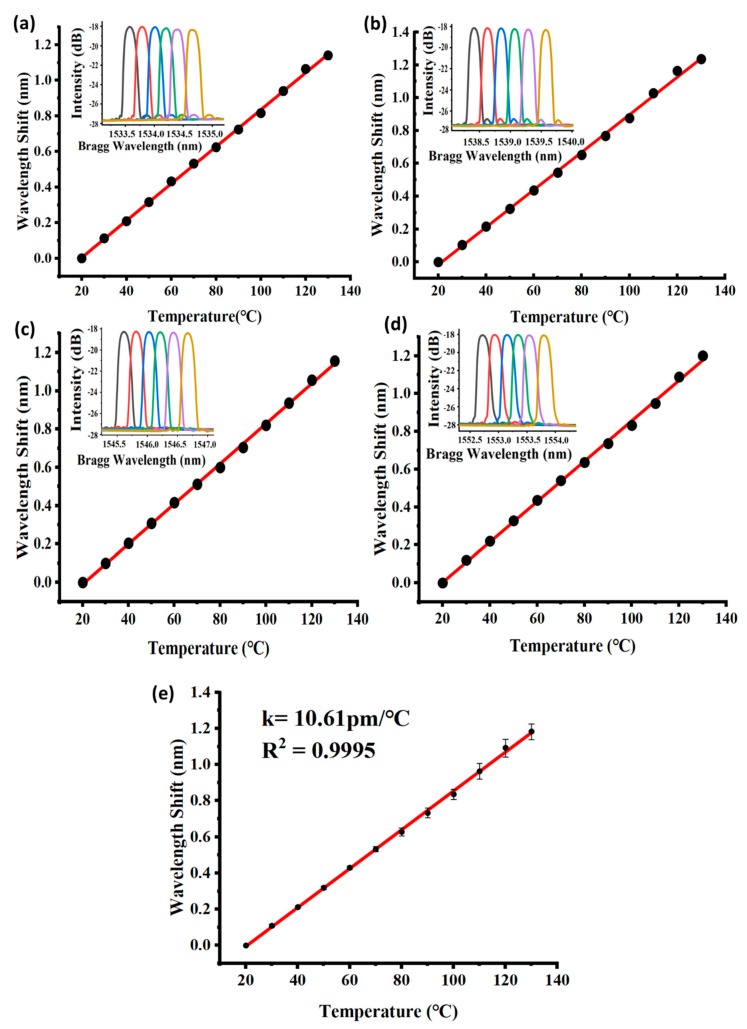
Central wavelength shift of the quasi-distributed FBG sensors: (**a**–**d**) redshift of FBGs 1–4, (**e**) integrated fitting curve of the temperature-monitoring system.

**Figure 4 micromachines-10-00866-f004:**
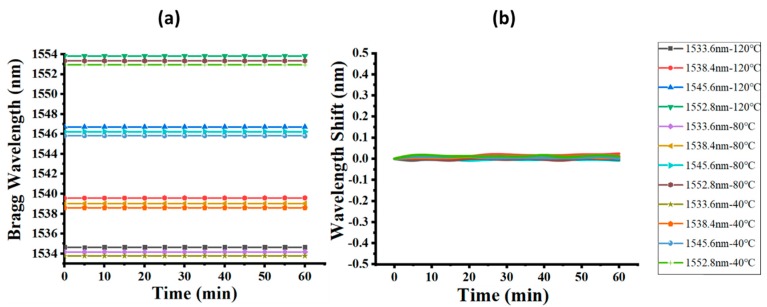
Stability test of the quasi-distributed FBG system: (**a**) dynamic Bragg wavelength recording under temperature setting in 1 h, (**b**) dynamic Bragg wavelength shift in 1 h under temperature setting.

**Figure 5 micromachines-10-00866-f005:**
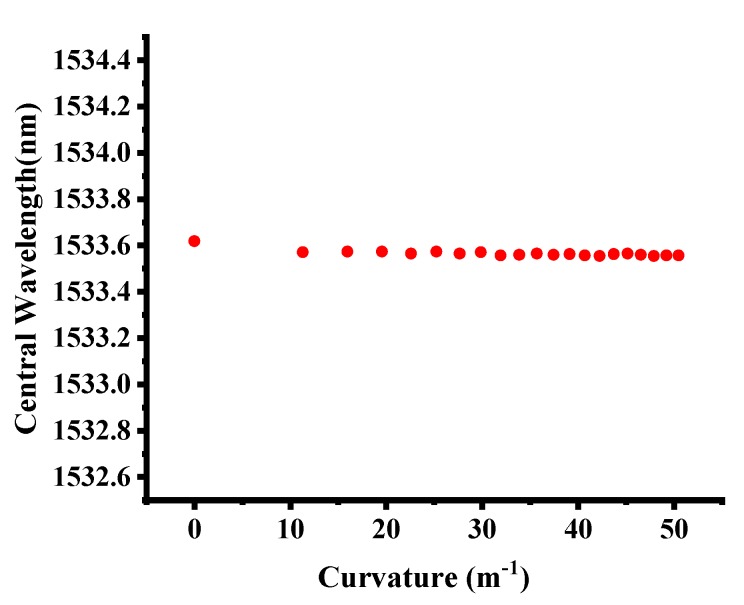
FBG central wavelength under different curvatures.

**Figure 6 micromachines-10-00866-f006:**
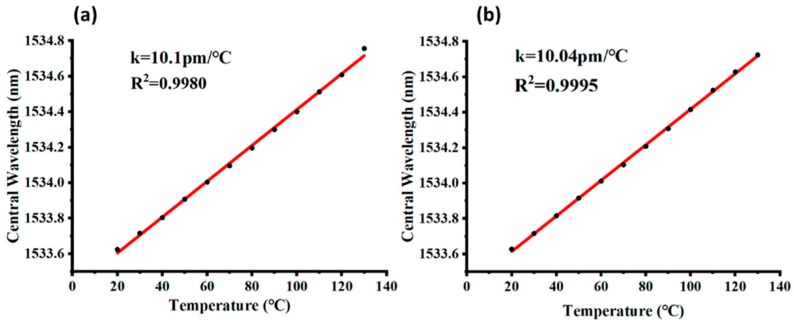
Sensitivity of the developed FBG-based fabric sensor: (**a**) between-wall integration method, (**b**) cross-wall integration method.

**Figure 7 micromachines-10-00866-f007:**
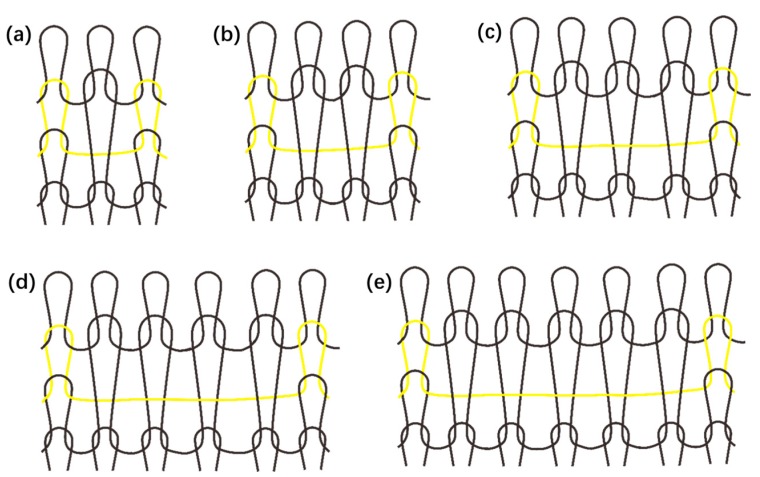
Miss/float stitch knitting structure design: (**a**) 1 × 1 float, (**b**) 1 × 2 float, (**c**) 1 × 3 float, (**d**) 1 × 4 float, (**e**) 1 × 5 float.

**Figure 8 micromachines-10-00866-f008:**
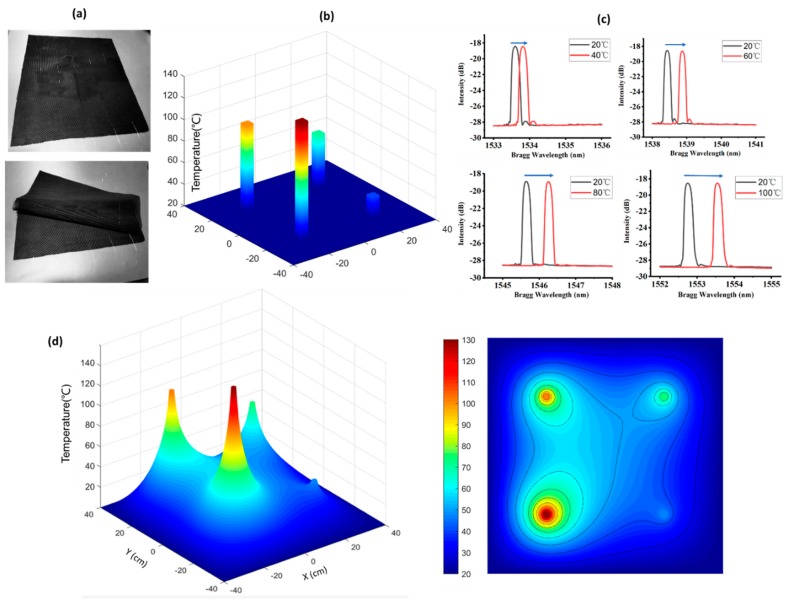
Experimental and simulation results of the developed temperature fabric sensor: (**a**) developed fabric sensor, (**b**) experimentally measured temperature distribution, (**c**) experimentally measured wavelength shift of four FBGs at different temperature conditions, (**d**) simulation results.

**Table 1 micromachines-10-00866-t001:** Temperature sensitivity results of each FBG in heating experiment.

Repeat No.-FBG No.	Sensitivity (pm/°C)	Linear Correlation Coefficient
1-1	11.00	0.9971
1-2	11.78	0.9944
1-3	10.64	0.9993
1-4	11.38	0.9970
2-1	10.38	0.9992
2-2	11.43	0.9983
2-3	10.47	0.9990
2-4	10.67	0.9987
3-1	10.83	0.9931
3-2	11.51	0.9865
3-3	10.78	0.9967
3-4	10.91	0.9946

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
