# Peer review of "Multifunctional Textile Platform for Fiber Optic Wearable Temperature-Monitoring Application"

_micromachines, 2019, doi:10.3390/mi10120866_

Round 1
Reviewer 1 Report
The paper presents a study of temperature detection from a FBG-based textile platform. The authors do a good job of reporting on the methodology and evaluation results. The work is technically correct and the proposed method may be of interest to the scientific community. However, the sensor concept is not novel and the readers may have trouble recognizing the strengths of the work and the usefulness of the results. For these reasons, authors should improve their manuscript, and the following comments should help them.
Major concerns:
First of all, authors should clearly explain why fibre-optic technology is of interest to them. There are many electronic temperature sensors. Which applications will the fibre-optic sensors be better for? Why will they be better? Small size, low weight, excellent multiplexing. One of many advantages of the fibre-optic technology is its insensitivity to electromagnetic radiation. This allows the use of fibre-optic sensors in the MRI environment. Please include the results shown in the latest literature [1] describing an MRI-compatible sensor. Another important issue is the sensitivity of FBGs both to temperature and strain. When talking on wearable textiles, the strain-induced Bragg wavelength shift cannot be ignored. Each movement causes an influence on the sensor and the spectral response of the sensor. How then will the proposed measurement method be effective? How sensitive is the measurement to body movements? Although this paper is probably not the right place for a complete characterisation of the effects of motion, some visual illustration of the effects of motion would be very useful for directing future work.
Other comments:
Abstract. Please, do not use trivial ‘the system demonstrated good sensitivity’. Use ‘the system demonstrated the sensitivity of…’. The first two sentences of the Introduction do not bring substantive content in terms of the manuscript subject. What equations (1)-(3) result from? Any references? A ‘thermal chamber’ sounds better than a ‘box’. In the context of half-loop or U-shape arrangements of optical fibre I found some works [2]-[5]. Please refer to the literature in the Discussion section. How are the authors going to interrogate the FBGs embedded into textiles? A good solution would be a small, lightweight interrogation system with wireless data transmission. It seems that the interest in electric and fibre-optic sensors embedded into textiles has decreased in recent years due to the need to have a wide range of T-shirt sizes to adjust to the body. These sensors are now being replaced by patch monitors [6] or sensors embedded into a bed or seat that do not require additional actions to prepare the patient for monitoring. Why the use of wearable textiles is still worthy of attention? Please explain this.
[1] L. Dziuda et al., A study of the relationship between the level of anxiety declared by MRI patients in the STAI questionnaire and their respiratory rate acquired by a fibre-optic sensor system, Sci. Rep. 9, 4341 (2019).
[2] A. T. Augousti, F.-X. Maletras, and J. Mason, “Improved fiber-optic respiratory monitoring using a figure-of-eight coil,” Physiol. Meas. 26(5), 585–590 (2005).
[3] J. De Jonckheere et al., OFSETH: smart medical textile for continuous monitoring of respiratory motions under magnetic resonance imaging, in Proc. of the 31st Annual Int. Conf. of the IEEE EMBS, 1473–1476 (2009).
[4] J. Witt et al., Smart medical textiles with embedded optical fibre sensors for continuous monitoring of respiratory movements during MRI, Proc. SPIE 7653, 76533B (2010).
[5] K. Krebber, S. Liehr, and J. Witt,“Smart technical textiles based on fiber-optic sensors, Proc. SPIE 8421, 84212A (2012).
[6] S. S. Lobodzinski, ECG patch monitors for assessment of cardiac rhythm abnormalities, Prog. Cardiovas. Dis., 56(2), 224-229 (2013).
Reviewer 2 Report
The authors report on a wearable temperature sensor based on FBGs. The research is interesting, and the paper is well written. Please find in the following a set of suggestions which may help to improve the quality of the paper.
Descriptions in Fig. 2 are too small. Please increase the font size. Please do the same in the inset plots in Fig. 3 and the descriptions of the curves in Fig. 4. Please add a comment on the expected cross-sensitivity if longitudinal strain is applied to the fiber. Results shown in Fig. 8 should be time-dependent. Please specify the time interval which was considered in the results. Also please add a comment on the time response of the sensor.Author Response
Please see the attachment.

Reviewer 3 Report
In this paper, the authors reported a multifunctional textile based on FBG for temperature monitoring. The FBF based quasi-distributed sensing system demonstrated good sensitivity with high stability in various temperature environments. Optical fiber for smart textiles is an interesting topic. The article is very well written, clear and concise and suitable for the scope of the journal. Only one suggestion: FBG in special fiber such as polymer optical fiber Bragg grating for thermal detect, multicore fiber Bragg grating for 3d shape sensing is suggested in the introduction part.
Round 2
Reviewer 1 Report
All the reviewer's comments have been included. Thank you.